# Covid-19 vaccine uptake and its associated factors among adult population in Dangila district, Awi Zone, Northwest Ethiopia: A mixed method study

**Girma Tadesse Wassie[1], Yeshambel Agumas Ambelie[2], Tsion Adebabay[3], Almaw Genet Yeshiwas[4], Eneyew Talie Fenta[5], Endeshaw Chekol Abebe[6], Gizachew Tadesse Wassie**[7], **Getachew Asmare Adella[8], Denekew Tenaw Anley**[9] *

1 Dangla Woreda Health Office, Dangla, Ethiopia, 2 Health System Leadership Director, Leadership Development Program, Department of Health System Management, School of Public Health, Bahir Dar University, Bahir Dar, Ethiopia, 3 School of Public Health, Bahir Dar University, Bahir Dar, Ethiopia, 4 Department of Environmental Health, College of Medicine and Health Sciences, Injibara University, Injibara, Ethiopia, 5 Department of Public Health, College of Medicine and Health Sciences, Injibara University, Injibara, Ethiopia, 6 Department of Medical Biochemistry, College of Health Sciences, Debre Tabor University, Debre Tabor, Ethiopia, 7 Department of Epidemiology and Biostatistics, School of Public Health, College of Medicine and Health Sciences, Bahir Dar University, Bahir Dar, Ethiopia, 8 Department of Reproductive Health and Nutrition, School of Public Health, Woliata Sodo University, Sodo, Ethiopia, 9 Department of Public Health, College of Health Sciences, Debre Tabor University, Debre Tabor, Ethiopia

* denekewtenaw7@gmail.com

**Data Availability Statement:** All relevant data are within the manuscript and its Supporting information files.

# Abstract

## Introduction

Vaccination is the most cost-effective approach that significantly reduces morbidity and mortality related to Coronavirus disease -19 (COVID-19). Nevertheless, there is a lack of information on the COVID-19 vaccine uptake and related factors in Ethiopia including the research area.

## Objective

To assess COVID-19 vaccine uptake and its associated factors among adult population in Dangila District, Awi Zone, Northwest Ethiopia, 2023.

## Methods

A community-based mixed-type study design was conducted from Oct, 15-Nov 15/2022. The study population was chosen using the multistage stratified random sampling technique for the quantitative study and the purposive sampling method for the qualitative inquiry. The collected data were managed and analyzed using SPSS version 25. Bivariable and multivariable logistic regressions were employed to identify factors associated with vaccine uptakes. In the qualitative part of the study, key informant interview was applied. After the interview was listened, the transcripts were coded and categorized into themes, and analyzed using Atlas.ti 7 software. Finally, the finding was triangulated with the quantitative results.

**Funding:** The author(s) received no specific funding for this work.

**Competing interests:** The authors have declared that no competing interests exist.

## Result

The vaccine uptake among the adult population was found to be 47% (95% CL: 42.7%, 51.0%). History of having test for COVID-19 (AOR: 1.70, 95% CI: 1.02, 2.84), good knowledge about COVID-19 vaccine (AOR; 3.12, 95% CI; 2.11, 4.59), no formal education (AOR: 1.78, 95%: 1.26, 2.58), good attitude (AOR: 3.21, 95% CI: 2.13, 4.89), being in poor Income category (AOR: 1.83, 95% CI: 1.08, 3.06), being female (AOR: 1.75, 95% CI: 1.2, 2.58) and living in rural area (AOR: 3.1, 95% CI: 1.87, 5.12) were significantly associated with vaccine uptake rate. The study also identified that misperceptions about the vaccine efficacy and safety, availability of vaccine, lack of knowledge about the vaccine, mistrust of the corona virus vaccine, fear of adverse effects, social media influence and religious beliefs were found to be barriers of COVID -19 vaccine uptake.

## Conclusion

In the Dangila district, adult population vaccination uptake for COVID-19 was comparatively low. To raise the rate of vaccination uptake, interventions must focus on the identified modifiable factors.

## Introduction

COVID-19, first emerged in the Wuhan province of China in December 2019, is caused by severe acute respiratory syndrome coronavirus-2 (SARS-CoV-2) [1–3]. It is a fatal viral disease that inflicts many countries around the world and becomes a major public health concern [4, 5]. Numerous COVID-19 preventive initiatives have been implemented by nations worldwide, including national lockdowns, quarantines, and movement restrictions. Actions have also been taken on an individual and group level to enhance hand hygiene, physically distance oneself from others, and use face masks [6]. In spite of these attempts to contain the worldwide pandemic, COVID-19 cases and newly discovered SARS-CoV-2 mutations are still rising [7]. A global COVID-19 vaccination program has been implemented in an effort to combat the pandemic and stop the spread of the disease [8].

Despite many attempts to contain its spread, it continues to become a significant public health concern and inflicts enormous burdens of morbidity and mortality while seriously challenging the globe to unprecedented strain on health system economies, and social life [9, 10].

COVID-19 vaccination is one of the most successful and cost-effective public health intervention strategies to mitigate the spread of SARS-CoV-2 and reduces the emergence of new strains [11–14]. Reports have shown that vaccination significantly reduces global morbidity and mortality related to COVID-19 [15, 16]. It is the means of blocking the transmission of the pandemic by building up herd immunity through large-scale vaccination [17, 18].

To date, more than 12 billion of do of COVID-19 vaccines have been provided to nearly 5 billion people worldwide [19]. In Ethiopia, the COVID-19 vaccination program commenced on 13 March 2021, with priority given to health professionals and the elderly. Whereas, mass vaccination targeting all Ethiopia people aged 12 years and above began on 16 November 2021 [20]. So far, 37.9 million of the Ethiopian population has been vaccinated with WHO-approved vaccines [21]. In Ethiopia, there have been 490,288 confirmed COVID 19 cases and 7,551 deaths reported [22]. Compared to the previous reports, this indicates a significant increment of the magnitude of COVID-19 morbidity and mortality. As a result of the high

spreading nature of the disease and its huge negative impact, vaccination becomes the top choice to control its transmission and to end the pandemic.

The rapid development of effective vaccines against COVID-19 represents an extraordinary achievement, but it also fuels vaccine hesitancy [23, 24]. Globally, there has been a rise in COVID-19 vaccine hesitancy. Low vaccine acceptance is evidenced from studies in different countries, such as Saudi Arabia (50%), Jordan (37%), the US (41%), Turkey (37%), and other large-scale studies involving 16 countries (52.0%) and Ethiopia (31.4) [25–30].

Vaccine uptake is being impeded by community vaccine hesitancy, a global issue that is becoming more prevalent. Numerous research have indicated that vaccine hesitation can stem from a variety of factors, such as concerns about vaccine safety, fear of contracting COVID-19, and fear of genetic consequences [31, 32].

Similarly, COVID 19 vaccine acceptance in Ethiopia ranged from 31.4% to 92.33% [33]. However, the previous studies conducted were on health care providers at health institution level, which pointed the need to conduct it at the community level. Besides, there was scarcity of information on COVID-19 vaccine uptake in different areas including Dangila district [34, 35]. So, this study was aimed at assessing COVID-19 vaccine up take and associated factors among adult population in Dangila district, Amhara region, Ethiopia.

## Materials and methods

### Study design and setting

A community level an explanatory sequential mixed-method study was employed. We preferred this approach to clarify or elaborate on quantitative findings by exploring participants' perspectives or contexts. The study was conducted from October 15 to November 15, 2022 in Dangila District of Awi Zone, North west Ethiopia. Dangila District is one of the woredas of Awi zone in Amhara regional state, containing 41 kebeles and estimated 204 thousand total populations. Dangila is the administrative town of this district, which is located 486 kilometers from Addis Ababa, the capital city of Ethiopia, and 78 kilometers far away from Bahir Dar, the capital city of Amhara regional state. The district has one hospital, seven health centers, and 41 health posts to provide a broad range of medical services for all age groups.

### Population

The source population was all adult individuals whose age was greater than 18 years and living in Dangila district. Whereas, the study population was all adult individuals whose age was greater than 18 years, and living in the selected kebeles of the study area. The participants were recruited from October 15 to November 15, 2022. For the qualitative study, community leaders, religious leaders, Health extension workers and EPI focal were the study participant.

### Eligibility criteria

For the quantitative component, adult population whose age was greater than 18 years living in the selected kebeles for at least the last 6 months preceding the study were included in the study. For the qualitative part any legally authorized or delegated person in the selected institutions were included. However, study participants who were unable to talk and refused the interview were excluded.

### Dependent variable

COVID-19 vaccine uptake (yes/no).

## Independent variables

**Socio-demographic characteristics**: age, sex, marital status, educational status, occupation, ethnicity, religion, income, family size, and residence.

**Health status-related characteristics**: personal history of COVID-19 infection, family history of COVID-19 infection, history of COVID-19 test, history of contact with confirmed COVID-19 patients, history of chronic diseases.

**Personal factor: Knowledge on** COVID 19 vaccine, attitude towards COVID-19 vaccine.

## Operational definition

**COVID-19 vaccine uptake**: Study participants who took the vaccine at least once (receipts of ≥1 COVID-19 vaccine dose) were categorized as vaccinated, whereas, those who didn't take at least 1 dose of COVID-19 vaccine were labeled as unvaccinated [36].

**Knowledge about the COVID 19 vaccine uptake:** Knowledge towards COVID-19 vaccine uptake was defined as understanding adverse effect, type of vaccine, safety of vaccine and vaccine accessibility. A respondent was considered as having good knowledge if he/she answered two-third and above of knowledge related questions. If otherwise, considered as having poor knowledge [37].

**Attitude about the COVID 19 vaccine up take:** Understanding and interpreting the essentiality of vaccine risk and trust of information from media sources as well as effectiveness of vaccine. Positive attitude: if the respondent answered two-third and above of attitude related questions. Negative attitude: if the respondent answered below two-third of attitude related questions [37].

## Sample size determination and sampling technique

The sample size was calculated using single population proportion formula by considering the following assumptions: 95% confidence interval, 80% power, the magnitude of vaccine uptake (P = 39.4%) taken from the previous study done in Hararghe, a zone in Ethiopia [6];

$$n = \frac{(Za/2)^2 p(1-p)}{d^2}$$

Where:

n = sample size

Z = **Standard** normal distribution corresponding to significance level at alpha value of 0.05 or Confidence interval (CI), 95% = 1.96

P = expected proportion (0.394) of adult populations of COVID -19 vaccine uptake

d = margin of error (5%) = 0.05

Therefore:

$$n = \frac{(1.96)^2 \times 0.394 \times (0.606)}{0.0025} = 367$$

The sample size was adjusted for design effect and non-response rate. Design effect of 1.5 was taken and the adjusted sample size was determined to be **550**. By adjusting it for 10% non-response rate, the final sample size became 605.

In the qualitative part, to identify barriers regarding vaccine uptake, we preferred to use key informants so that the quantitative finding could be supported for better recommendation and intervention. Hence, sample size was determined based on the information saturation of responses.

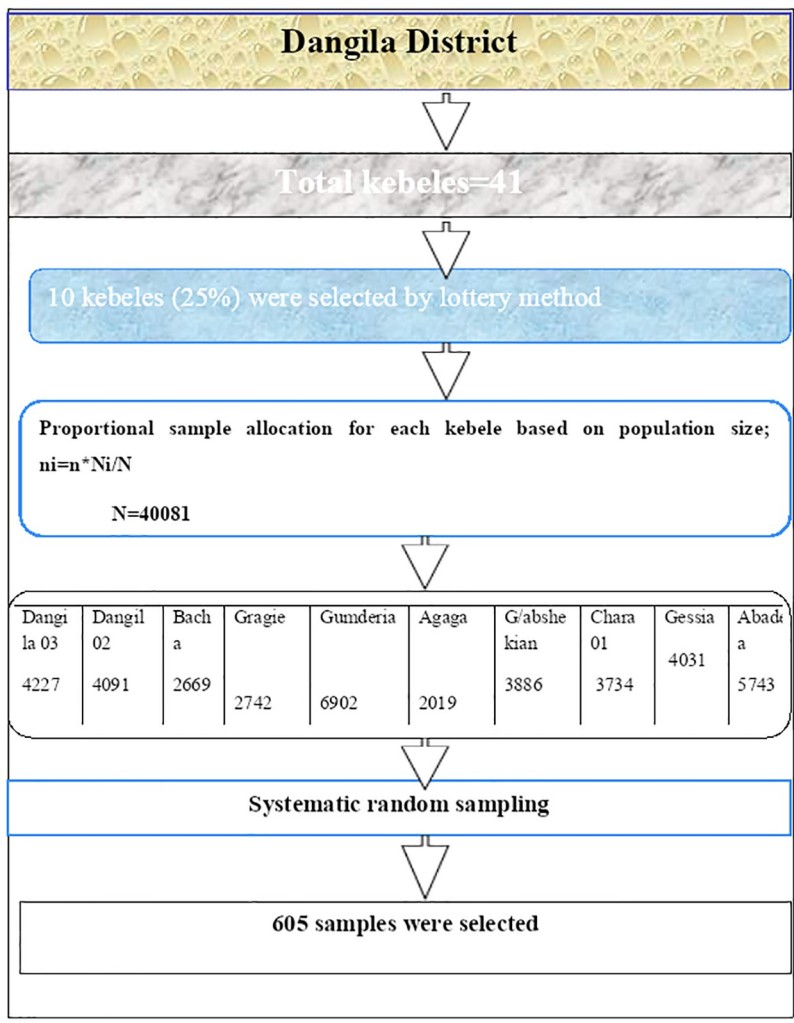

**Fig 1. Schematic representation of the sampling procedure.**

Multistage stratified random sampling technique was applied to select the study population. Lottery method was first used to select ten kebeles (namely Kebele 02, Kebele 03, Bacha, Gerargie, Agaga, Gundry, Abdra, Chara 01, Gesia 01, and Gult out of 41 kebeles in Dangila District. The sampling procedure is illustrated by the following diagram (Fig 1).

For the qualitative assessment part, individuals who have first-hand knowledge about the community, HEWs, kebele leaders, religions leader, EPI focal persons, and head of city administration health office were selected purposively as a key-informants.

## Data collection tools and techniques

Quantitative data were collected by the face-to-face interview method using the pretested structured questionnaire. The questionnaire was comprised of socio-demographic variables, health related variables, and COVID -19 vaccine related variables. Qualitative data was collected using key informant interview by using 7 open ended Interview guides. Probing questions were asked based on participants' response.

## Data quality assurance

The questionnaire prepared in English was translated to the local language, Amharic, and then retranslated back to the English to ensure consistency. A questionnaire was reviewed by a panel of experts for construct and content validity. Moreover, pretesting was carried out among 5% of the research participants in the Afsa cluster kebeles prior to the actual data collection period to determine the validity and reliability of the questionnaire and to look at practical concerns in participant selection. The reliability was validated using Cronbach's alpha test for all questions and the result was found to be 0.749. Before the commencement of the real data collection, the necessary improvements, such as phrasing, logical sequencing, correcting inconsistencies, and faults in the skip patterns were made. Besides, training was given to the data collectors and supervisors on the aim of the study, the content of the tools, confidentiality, and informed consent for two days. The investigators ensured that the study was trustworthy based on Lincoln and Cuba's criteria of credibility, dependability, conformability, and transferability. To maintain the credibility of the research findings, the study participants were observed persistently at the time of the interview. Peer-debriefing was done for the questioner. Dependability was attained through accurate documentation by minimizing spelling errors through frequently observing data and including all documents in the final report, such as including the notes written during the interview and ensuring that the details of the procedure was described to allow the readers to see the basis upon which conclusions were made. To achieve conformability of the study findings, raw data, interview and observation notes, documents and records collected from the field, and others were documented for cross checking. To maintain the transferability of the finding, appropriate probes was used to obtain detailed information on responses, and study participants were selected based on their specific purpose to answer study questions and to get greater in-depth findings.

## Ethics approval and consent to participate

The Institutional Review Board (IRB) of the Bahir Dar University granted ethical approval with the reference number 577/2022. Besides, a written collaboration letter was obtained from Dangila District Health office. Informed consent was obtained from each selected study participant. Confidentiality of information and anonymity of the data collection procedure were ensured. Authors hadn't access to information that could identify individual participants Moreover, during and after the data collection process, appropriate infection prevention measures and principles related to COVID-19 were strictly kept.

## Data management and analysis

Both data collection and entry were done by kobo toolbox simultaneously. The collected data were exported to SPSS version 25.0 for data processing and analysis. Characteristics of the study participants were analyzed using descriptive statistics such as frequency, percentage, and were presented using tables and charts. Association between dependent and independent variables was identified using logistic regression analysis. Model fitness was assessed using Hosmer–Lemeshow test. Variables with p-value of $\leq 0.25$ on bivariable analysis were selected for multivariable analysis. In the multivariable model, Adjusted Odds Ratio (AOR) with 95% CI was used to identify independent predictors of COVID-19 vaccine uptake. Statistically significant association was declared at p-value $<0.05$.

For qualitative part, the principal investigator transcribed verbatim the audio-recorded data word by word on participants' local language. A total of 66 codes were made after repetitive readings of the transcription. Combining similar meanings, 23 categories were formed; from

there, 7 sub-themes and 3 major themes were identified. Atlas.it 7 software was used to conduct the analysis using a thematic approach.

## Results

### Socio-demographic characteristics

Six hundred respondents had given a complete with a response rate of 99.1%. Nearly, 51% of respondents were males. 482 (72.5%) of respondent's marital status was married. 567 (94.5%) respondents were orthodox Christian followers. The mean (±SD) age of respondents was 38.34 (±15.2) years, respectively. Regarding their educational status, 284 (47.3%) had no formal education. Of the total participants, 469(78.2%) lived in <5 family size house. There were five key informant interviews including the head of the city administration health office, health extension worker, immunization officer nurse, religious leader and the head of kebele (Table 1).

### Health related characteristics

Only 19(3.2%) of respondents had history COVID-19, and 112(22.3%) of respondents were tested for COV-19 (Table 2).

### Respondents' knowledge and attitude characteristics

About 290(48.3%) of respondents were found to have good knowledge about COVID-19 vaccine uptake.

Only 225(35.7%) of respondents were found to have positive attitude towards COVID-19 vaccine. Besides, 365(60.8%) respondents believe that COVID vaccines are not safe for health (Table 3). The item by item knowledge and attitude related characteristics of study subjects are presented in S1 and S2 Tables respectively.

### Respondents COVID -19 vaccine uptake characteristics

About 282 (47%) respondents were vaccinated with any of the COVID-19 vaccines at least one dose, whereas about 318(53%) were not vaccinated or had not taken the vaccine. The reason not vaccinated were fear of adverse effect 70(11.7%) (Table 4).

### Prevalence of COVID -19 vaccine uptakes

The prevalence of vaccine uptake rate in Dangila district was 47% [95% CL: 42.7%, 51.0%] (Fig 2).

### Reasons for not taking COVID -19 vaccinations

A number of reasons for not taking COVID-19 vaccines were identified, and illustrated by the following figure (Fig 3).

### Factors associated with vaccine uptake

During the bivariable logistic regression analysis; income of household, knowledge of respondents, attitude respondents, residence, educational status, sex, test of COVID -19, family size, age of respondent were candidate (p≤0.25) for multivariable analysis.

In multivariable logistic regression analysis variables like income of house hold, knowledge, attitude, residence, educational status, test COVID -19 and sex, were found to be significantly associated with the vaccine uptake rate of individuals.

**Table 1. Socio-economic and demographic characteristics of respondents in Awi Zone, Dangila district, Northwest Ethiopia, 2023(n = 600).**

| Characteristics | Frequency(n = 600) | Percentage % |
|---|---|---|
| **Marital status** | | |
| Single | 96 | 16 |
| Married | 451 | 72.5 |
| Divorced | 36 | 6 |
| Widowed | 17 | 2.8 |
| **Education status of respondents** | | |
| no formal education | 284 | 47.3 |
| primary education | 124 | 20.7 |
| secondary & above | 192 | 32 |
| **Residence** | | |
| Urban | 127 | 21.2 |
| Rural | 473 | 78.8 |
| **Sex of respondents** | | |
| Male | 303 | 50.5 |
| Female | 297 | 49.5 |
| **Age of respondents** | | |
| 18–29 | 201 | 33.5 |
| 30–39 | 160 | 26.7 |
| 40–49 | 117 | 19.5 |
| > = 50 | 122 | 20.3 |
| **Income of respondent** | | |
| Poor | 265 | 44.2 |
| Medium | 183 | 32.2 |
| Rich | 142 | 23.7 |
| **Religious of Respondent** | | |
| Orthodox | 567 | 84.5 |
| Muslim | 33 | 5.5 |
| **Family size Respondent** | | |
| <5 | 469 | 78.2 |
| > = 5 | 131 | 21.8 |
| **Occupational status of Respondent** | | |
| House wife | 106 | 17.7 |
| Merchant | 117 | 19.5 |
| Farmer | 225 | 37.5 |
| Gov,t employee | 47 | 7.8 |
| Private employee | 14 | 2.3 |
| Student | 40 | 6.7 |
| Daily labour | 31 | 5.2 |
| Other | 20 | 3.3 |

The odds of vaccine uptake rate among respondents who had good knowledge about Covid-19 disease were 3.1times (AOR = 3.12, 95% CI: (2.11, 4.59) higher uptake of COVID-19 vaccines than poor knowledge individuals. The knowledge aspect of the community was also assessed through key informant interview. Most of the study participants reported that the community has no proper understanding of COVID 19 disease or vaccine. This lack of awareness about the vaccine is caused by poor social mobilization works by the concerned bodies.

**Table 2. Health-related characteristics of respondents in Awi zone, Dangila district, Northwest Ethiopia, 2023 (n = 600).**

| Characteristics | Frequency(n = 600) | Percentage % |
|---|---|---|
| History of COVID-19 | | |
| No | 581 | 96.8 |
| Yes | 19 | 3.2 |
| Family history of COVID 19 | | |
| No | 585 | 97 |
| Yes | 18 | 3 |
| Ever tested for COVID-19 | | |
| No | 488 | 77.7 |
| Yes | 112 | 22.3 |
| Test result | | |
| Positive | 19 | 18.3 |
| Negative | 85 | 81.7 |
| History of contact with confirmed COVID-19 | | |
| No | 586 | 97.7 |
| Yes | 14 | 2.3 |
| Chronic disease | | |
| No | 520 | 86.7 |
| Yes | 80 | 13.3 |

*"The leaders of the community are not committed to receiving the vaccine and acting as role model, in addition, there are misunderstandings and incorrect ideas in the community that are linked to their religion and the vaccine's possibility to induce blood clotting". (A 50 year old, male head of health office). "The main issues we faced include lack of community knowledge, a short-age of vaccines on schedule, false information on social media, and a lack of commitment from leaders to set an example by having the vaccine first". (A 28 years old female health extension worker).*

Similarly, respondents who had positive attitude about Covid-19 vaccine uptake were 3.2times (AOR = 3.21, 95% CI: (2.13, 4.97) higher uptake the COVID vaccine than Negative attitude individuals. It was also supported by the qualitative finding. Due to an inadequate understanding and poor attitude towards the COVID-19 vaccine, the study's participants indicated that the majority of the population lacks confidence in the vaccine and on political leaders. *"The majority of people were vaccine-hesitant due to a lack of adequate information; hence we were unable to advocate vaccination throughout the entire community". (A 50 years old male head of health office).*

**Table 3. Knowledge, attitude characteristics of respondents in Awi Zone, Dangila district, Northwest Ethiopia, 2023(n = 600).**

| Characteristics | Frequency(n = 600) | Percentage |
|---|---|---|
| **Knowledge status of respondents** | | |
| Poor | 310 | 51.7 |
| Good | 290 | 48.3 |
| **Attitude status of respondents** | | |
| Positive | 225 | 37.5 |
| Negative | 375 | 62.5 |

**Table 4. COVID -19 vaccine uptake characteristics of respondents in Awi Zone, Dangila district, Northwest Ethiopia, 2023(n = 600).**

| Characteristics | Frequency(n = 600) | Percentage % |
|---|---|---|
| Vaccines status | | |
| No | 318 | 53 |
| Yes | 282 | 47 |
| **Types vaccine vaccinated** | | |
| Pfizer | 24 | 4 |
| Johnson & Johnson | 184 | 30.7 |
| Astrazeca | **72** | **12** |
| Other | **2** | **0.3** |
| *Vaccine dose* | | |
| One | 186 | 31 |
| Two | 94 | 15.7 |
| Full dose | 8 | 1.3 |
| **Reason not vaccinated(318)** | | |
| Fear of injection | 44 | 7.3 |
| Fear of adverse effects | 70 | 11.7 |
| Lack of vaccine access | 15 | 2.5 |
| Doubts on vaccine efficacy | 38 | 6.3 |
| Because of my religion | 38 | 6.3 |
| Personal health status | 28 | 4.3 |
| I prefer other protection | 39 | 6.5 |
| Other- | 46 | 8 |

**other-absenteeism

-political issue

-COVID-19 is not Health problem

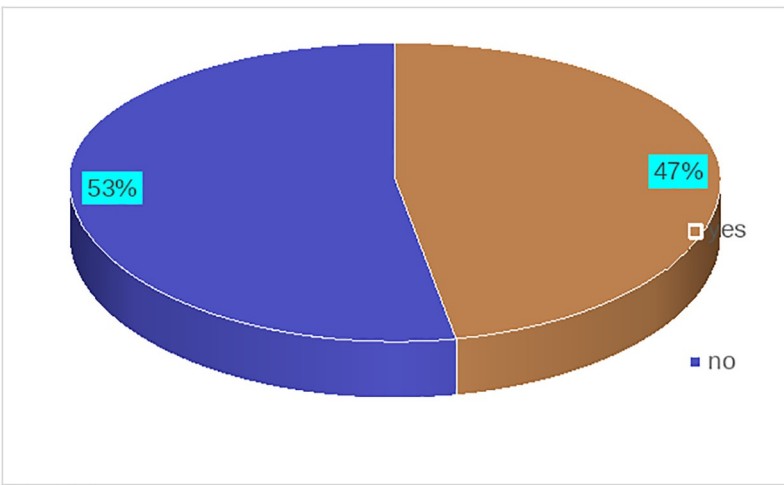

**Fig 2. Prevalence of COVID-19 vaccine uptake among respondents in Awi Zone, Dangila district, Northwest Ethiopia, 2023.**

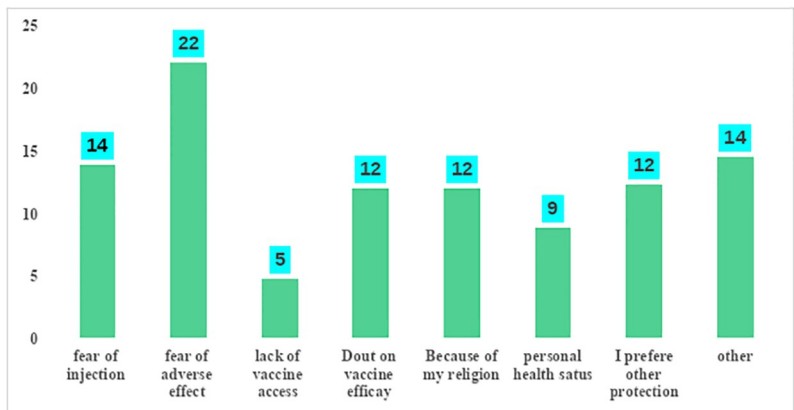

**Fig 3. Reason for not taking vaccination among respondents in Awi Zone, Dangila district, Northwest Ethiopia, 2023(n = 318).** **others:—absenteeism,—Political issue,—COVID -19 is not my problem.

Besides, the participants discussed how their refusal to receive the vaccine was a result of their concern over adverse effects such as headache, fever, and joint pain. They claimed that the people are afraid of blood clotting, infertility, its impact on their religion, and its possibility of causing death. *"Due to false information, people are afraid to get the vaccine because they think it may cause adverse effects like headaches, fevers, and joint discomfort." (A 50 years old male Islamic religion leader).*

*"There was misinformation disseminated saying that vaccinations can cause death, that may alter our religious beliefs, and that they may induce side effects like fever and headache."*

*(A 38 years old male kebele leader).*

The majority of the population does not understand the COVID-19 vaccination and believes that it may not be effective in stopping the disease and may even have unexpected side effects. *". . . . . . . . . . . . . . . . . . but the public has expressed a variety of unfavorable opinions, believing that vaccines may be harmful to their health. Additionally, they hold unfavorable opinions and think the vaccine might be connected to their religion, complaints about the effectiveness and adverse effects of vaccinations have been raised". (A 50-year male head of health office).* The participants agreed that it might take some time to raise awareness and influence community attitudes with continued social mobilization efforts.

*"I am confident in the vaccine's safety because it has been properly investigated, but because it is new, the community has not yet accepted it. In our experience, when other programs, like family planning hormonal contraception depo, first launched, we had trouble providing them, but as public awareness grew, more and more people started to take and prefer the vaccine".*

*(A 28-year female health extension worker).*

Besides, environmental factors were also found to affect the COVID 19 vaccine uptake. The study participants examined that social media misinformation affects public uptake of vaccines by providing false knowledge and misunderstanding on the COVID-19 vaccine.

*"Vaccine uptake was low, only some people volunteered to take the vaccine, majority of the people didn't take the vaccine due to misinformation dissemination on Facebook and other media which falsely let them be vaccine-hesitant"*

*(a 28 years old male nurse).*

*"The majority of people claim that there is no COVID-19 disease, and as a result, they were hesitant to get the vaccine because of vaccine misinformation on social media. Others argue that since there haven't been any deaths in our community, COVID-19 is no longer a disease."*

*(A 38 years old male kebele leader).*

Furthermore, the majority of responders stated that the community believes that the COVID-19 vaccine may have an impact on their religion.

*"The majority of the population did not voluntarily receive vaccines. They are ignorant of the vaccination's efficacy and associate it with religion, saying things like, "God knows my health, and I didn't take the vaccine, which could result in death and infertility." (A 28 years old male nurse). "I believe that the COVID-19 illness is avoidable with vaccination and hand washing, but the majority of individuals refuse to get the vaccine because, in their thoughts, they trust in God. Given that the illness is fatal, we must get the vaccine to prevent it*

*(A 50 years old male Islamic religion leader).*

Additionally, Respondents who had no formal education were 1.7times (AOR = 1.78, 95%: (1.07, 2.97) higher uptake of COVID-19 vaccines than counterparts (Table 5).

The following table shows the qualitatively identified barriers of COVID 19 vaccine acceptance, classified as three main themes with seven sub-themes (Table 6).

## Discussion

The government of Ethiopia in collaboration with different partners has implemented different types of interventions to alleviate the prevalence of COVID-19 disease. Periodic monitoring and evaluation of the prevalence of COVID-19 disease and vaccine uptake rate are important for public health action for evaluating the implemented intervention programs. According to the findings of the current study, the prevalence of COVID-19 vaccine uptake rate of individuals is still lower.

In the same way, the finding of this study is also higher than the previously conducted surveys done at Hararghe which showed that about 39.4% of participants had taken the vaccine [6], and the vaccine acceptance is 21% in Egypt [38]. The discrepancies might be due to differences in the community awareness creation through mass media and health promotion, as well as the study conducted after 3[rd] round national COVID-19 vaccination campaign is done. Further, more in this study, near half of the participants had good knowledge towards COVID-19 vaccine, which is higher than the findings of previous studies.

The finding of this study was lower than the previous study conducted in different parts of the country and abroad. The finding is lower than the study done in Texas United States, France, and the United Kingdom 95.1%, 69%, and 82.5% COVID-19 vaccination coverage respectively [39–41]. The possible reason might to due to the awareness among the populations, more access to information in developed countries, and the availability of adequate resources (vaccines, facilities, and health personnel). The other possible reason is the late arrival of vaccines in developing countries including Ethiopia. Similarly, the study is lower

**Table 5. Factors associated with COVID -19 vaccine uptake among adult population at Dangila district, Awi Zone, northwest Ethiopia, 2023 (n = 600).**

| Characteristics | Vaccine uptake (n = 600) | | | | P-value |
|---|---|---|---|---|---|
| | Yes | No | COR (95% CL) | AOR (95% CL) | |
| **Income respondent** | | | | | |
| Poor | 125 | 140 | 1.37 (0.91, 2.07) | 1.83 (1.09, 3.08) | **0.024** |
| Medium | 101 | 92 | 1.69 (1.08, 2.62) | 0.98 (0.59, 1.63) | 0.59 |
| Rich | 56 | 86 | 1 | 1 | |
| **Knowledge** | | | | | |
| Good | 182 | 109 | 3.49 (2.49, 4.88) | 3.12 (2.11, 4.59) | **0.001** |
| Poor | 100 | 209 | 1 | 1 | |
| **Attitude** | | | | | |
| Positive | 147 | 80 | 3.24 (2.23, 4.57) | 3.21 (2.13, 4.97) | **0.001** |
| Negative | 135 | 238 | 1 | 1 | |
| **Residence** | | | | | |
| Rural | 242 | 231 | 2.28 (1.50, 3.45) | 3.10 (1.87, 5.12) | **0.01** |
| Urban | 40 | 87 | 1 | 1 | |
| **Educational status** | | | | | |
| No formal education | 152 | 132 | 1.68 (1.16, 2.43) | 1.78 (1. 07, 2.97) | **0.026** |
| Primary education | 52 | 72 | 1.05 (0.66, 1.67) | 1.31 (0.75, 2.26) | 0.331 |
| Secondary & above | 78 | 114 | 1 | 1 | |
| **Sex of respondents** | | | | | |
| Female | 152 | 145 | 1.40 (1.01, 1.924) | 1.75 (1.26, 2.56) | **0.03** |
| Male | 130 | 173 | 1 | | |
| **Test of COVID-19** | | | | | |
| Yes | 74 | 38 | 2.62 (1.71, 4.03) | 1.70 (1.02, 2.84) | **0.04** |
| No | 208 | 280 | 1 | | |
| **Family size** | | | | | |
| > = 5 | 69 | 62 | 1.32 (0.91, 1.97) | 1.04 (0.65,1.64) | 0.86 |
| <5 | 213 | 256 | 1 | | |
| **Age respondent** | | | | | |
| 18–29 | 95 | 106 | 0.78 (0.50, 1.16) | | |
| 30–39 | 65 | 95 | 0.6 (0.373, 1.965) | | |
| 40–49 | 57 | 60 | 0.83 (0.501, 1.233) | | |
| > = 50 | 65 | 57 | 1 | | |

**Table 6. Thematic findings on barriers of COVID -19 vaccine acceptance among adult population at Dangila district, Awi Zone, Northwest Ethiopia, 2023.**

| S. N | Main Theme | Sub-theme |
|---|---|---|
| 1 | Vaccine-related factors | • Availability of vaccine |
| | | • Misperceptions about the vaccine efficacy and safety |
| 2 | Personal factors | • Lack of knowledge about the COVID-19 vaccine |
| | | • Mis Trust |
| | | • Fear of adverse effect |
| 3 | Environmental factors | • Social media influence |
| | | • Religious beliefs |

than the study conducted in Ethiopia reporting that about 62.1% of them had received at least one dose of the COVID-19 vaccine [42]. This might be due to differences in socio-demographic factors, and time variation [43].

In this study analysis, different factors have an association with the vaccine uptake rate. Namely, Income of house hold, knowledge, attitude, residence, educational status, test COVID -19 and sex, religion, misinformation on social media, fear of side effects, mistrust of the vaccine, misperceptions about the vaccine efficacy and safety, availability of the vaccine, are found to be significantly associated with the vaccine uptake rate of individuals.

In this study, most of the study participants reported that the community has no proper understanding of COVID 19 disease or vaccine. It is reported that individuals those who had good knowledge and attitude were more likely than individuals those who had poor knowledge and attitude to accept the COVID-19 vaccine if it was available. The finding is consistent with previous studies done in Debre Tabor [44], Dessie [45]and Athens [46].

This study revealed that, out of the total participants, females (54%) had more vaccine coverage than males (46%). This finding is consistent with the findings reported from the United States and France, where more females were vaccinated than males [40, 47]. The possible reason for this discrepancy might be due to a difference in gender equality, equity and women empowerment and access to information between developed and developing countries including in Ethiopian. The other possible reason might be in developing countries including Ethiopia females are more responsible for child care. In this study indicated that the community believes that the COVID-19 vaccine may have an impact on their religion. This is consistent with the study in USA showed that religion was very important to them so that they would only accept the COVID-19 vaccine if provided enough information about it [48].

This study revealed, people who had no schooling were more likely to be vaccinated than people who had attended above-secondary school. This finding is consistent with the findings reported from Hararghe [6]. This could be due to the fact that uneducated people are more likely to accept the recommended vaccine without contemplating the potential side effects of the vaccine, while the educated could have more misconceptions, negative attitudes, and misuse of social media as their primary source of information.

This study revealed that, out of the total participants, rural residents (79%) had more vaccine coverage than urban (21%). This finding is inconsistent with the findings reported from Bangladesh that rural residents were more reluctant to vaccinate than those living in urban areas [49]. This could be due to the fact that rural living areas are more likely to uptake the recommended vaccine without considering the potential side effects of the vaccine, while urban residents could have more misconceptions, negative attitudes, and misuse of social media as their primary source of information.

This study also revealed that individuals those who had a test of COVID-19 and poor income of house household were more likely to uptake covid-19 vaccine than the counter parts. This might be if an individual had a fear of the disease, had positive attitude for finding or getting of the vaccination, but the person not give attention for the disease is resistance of the COVID-19 vaccines. On other hand those individuals perceived that, if they attacked by COVID-19, they cannot afford Medical expense.

The quantitative findings were further validated through a meticulous analysis of the qualitative data. Thematic exploration concerning barriers to COVID-19 vaccine acceptance among the adult population in Dangila district, Awi Zone, revealed three primary themes, each with seven sub-themes. These included Vaccine-related factors, Personal factors, and Environmental factors, encompassing issues such as the Availability of vaccine, Misperceptions about vaccine efficacy and safety, Lack of knowledge about the COVID-19 vaccine, Mis trust, Fear of adverse effects, Social media influence, and Religious beliefs. Notably, these

qualitative insights mirror the patterns observed in the quantitative analysis, affirming the robustness and consistency of the findings across both methodologies.

In this study use of social media misinformation was the main barrier of COVID vaccine uptake. This is similar with study done Nigeria, Gahana and Jordan which reported that lack of fact information about vaccine, mistrust and negative perceptions, and lack of trust on the vaccine were barriers of COVID -19 vaccine uptake respectively [50–52]. This study reported that fear of side effects like headache fever, headache and joint pain were the main factor for COVID- 19 vaccine hesitancy. The finding was consistent with the study done in Tehran stated that the main obstacles to vaccine acceptance were lack of belief in the vaccine's high efficacy, fear of its adverse effects [53].

In this study misperceptions about the vaccine efficacy and safety were the main barrier of COVID-19 vaccine uptake. This was similar with the study finding in Australia reported that misperception and misbeliefs developed from misinformation on social media were the berries of vaccine acceptance [54]. And the study Canada revealed that increased misperceptions are associated with greater vaccine hesitancy [55]. The study also indicated that vaccine availability was one factor of vaccine uptake according to the schedule. This finding was similar with the study in USA showed that lack of vaccines decreased participants' feeling of urgency, which in turn decreased their readiness and intention to receive vaccinations [56]. Generally, this study has come up with the community level evidences on factors which contribute to COVID-19 vaccine uptake, hence could be generalized. The mixed method approach employed could make the evidence more strong and timely. However, the study was not without limitations. The cross-sectional nature of the study doesn't allow making a temporal relationship between variables. Besides, potential recall bias might be there during asking past histories regarding to COVID-19.

## Conclusion

In this study COVID-19 vaccine uptake among adult population in Dangila district was lower. Being female, having a person with no formal education, tested for COVID-19, living a rural area, poor income household, having a good knowledge and having a good attitude were found to be significantly associated with COVID-19 vaccine uptake. This study also indicated that misperceptions about the vaccine efficacy and safety, availability of vaccine, lack of knowledge about the COVID-19 vaccine, mistrust on the vaccine, fear of adverse effect, social media influence and religious beliefs were barriers in COVID-19 vaccine uptake.

## Supporting information

**S1 File. The minimal anonymized dataset.**
(CSV)

**S1 Table. COVID-19 related knowledge of study subjects.**
(DOCX)

**S2 Table. Attitude of study subjects towards COVID-19 vaccine.**
(DOCX)

## Acknowledgments

We appreciate the permission and support of the Bahir Dar University and Dangila district Health Office. We also want to thank the data collectors and supervisors for their role in the accomplishment of this study.

## Author Contributions

**Conceptualization:** Girma Tadesse Wassie, Yeshambel Agumas Ambelie, Tsion Adebabay, Almaw Genet Yeshiwas, Endeshaw Chekol Abebe, Gizachew Tadesse Wassie, Getachew Asmare Adella, Denekew Tenaw Anley.

**Data curation:** Girma Tadesse Wassie, Yeshambel Agumas Ambelie, Tsion Adebabay, Almaw Genet Yeshiwas, Gizachew Tadesse Wassie, Getachew Asmare Adella, Denekew Tenaw Anley.

**Formal analysis:** Girma Tadesse Wassie, Tsion Adebabay, Almaw Genet Yeshiwas, Eneyew Talie Fenta, Endeshaw Chekol Abebe, Denekew Tenaw Anley.

**Investigation:** Girma Tadesse Wassie, Yeshambel Agumas Ambelie.

**Methodology:** Eneyew Talie Fenta, Denekew Tenaw Anley.

**Software:** Tsion Adebabay.

**Supervision:** Yeshambel Agumas Ambelie, Tsion Adebabay, Getachew Asmare Adella.

**Validation:** Yeshambel Agumas Ambelie, Gizachew Tadesse Wassie.

**Visualization:** Eneyew Talie Fenta, Endeshaw Chekol Abebe, Gizachew Tadesse Wassie.

**Writing – review & editing:** Almaw Genet Yeshiwas, Endeshaw Chekol Abebe, Gizachew Tadesse Wassie, Getachew Asmare Adella.

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
