## [Decision Letter · Decision Letter 0]

20 Jul 2023

PONE-D-23-08920Covid-19 vaccine uptake and its associated factors among adult population in Dangila district, Awi Zone, Northwest Ethiopia: A semi-qualitative cross sectional studyPLOS ONE

Dear Dr. Anley,

Thank you for submitting your manuscript to PLOS ONE. After careful consideration, we feel that it has merit but does not fully meet PLOS ONE’s publication criteria as it currently stands. Therefore, we invite you to submit a revised version of the manuscript that addresses the points raised during the review process.

We look forward to receiving your revised manuscript.

Kind regards,

Mesfin Gebrehiwot Damtew

Academic Editor

PLOS ONE

Journal Requirements:

- https://doi.org/10.1177/20503121221077585

- https://doi.org/10.2147/IDR.S360103

- https://doi.org/10.3389/fpubh.2022.919494

In your revision ensure you cite all your sources (including your own works), and quote or rephrase any duplicated text outside the methods section. Further consideration is dependent on these concerns being addressed.

3. We note that Figure 1 in your submission contain map/satellite images which may be copyrighted. All PLOS content is published under the Creative Commons Attribution License (CC BY 4.0), which means that the manuscript, images, and Supporting Information files will be freely available online, and any third party is permitted to access, download, copy, distribute, and use these materials in any way, even commercially, with proper attribution. For these reasons, we cannot publish previously copyrighted maps or satellite images created using proprietary data, such as Google software (Google Maps, Street View, and Earth). For more information, see our copyright guidelines: http://journals.plos.org/plosone/s/licenses-and-copyright.

4. We are unable to open your Supporting Information file S1 File.sav. Please kindly revise as necessary and re-upload.

Reviewers' comments:

Reviewer's Responses to Questions

**Comments to the Author**

1. Is the manuscript technically sound, and do the data support the conclusions?

Reviewer #1: Yes

Reviewer #2: Partly

2. Has the statistical analysis been performed appropriately and rigorously? 

Reviewer #1: Yes

Reviewer #2: N/A

3. Have the authors made all data underlying the findings in their manuscript fully available?

Reviewer #1: No

Reviewer #2: Yes

4. Is the manuscript presented in an intelligible fashion and written in standard English?

Reviewer #1: Yes

Reviewer #2: No

5. Review Comments to the Author

Reviewer #1: Thank you this intersting piece of work on covid vaccine uptake from Ethiopia

The following key comments:

1. Integration of methods, data, findings ans conclusions is a key challenge of this work. The authors faced a great dealnof challenge to tringulate the qualitative and quantitative methodd. The findings of both methods should be mixed as appropriate instead of being put as separate pieces. So, please integrate the finding. This would pave ways to draw appropriate conclusion and and improve assimilation of findings

2. Qualitative methods seem overlooked. First ly, though saturation is mentioned, there is no adequate internal evidence that saturation is reached. 1. Tentative sample sizes were not mentioned, 2. Data triangulation is not well indicated. Secondly, data analysis and rigor is not well stated. What type of thematic analysis and coding was applied- open or priori codinf? The procedure paased through transcription, coding, and analysis overly depended on one person, the principal investigator. That is not auditable, not based on peer debriefying etc. So please mention the roles of othet authors in the qualiative process. Please indicate the extent to which subjective neitrality, refxivity, tick description, etc was ensured toward dependebality of the interpretations. Please revise and improve this aspect of the study.

3. In the abstract yoi mentioned..." trascribed reduced, coded, ......". This should be revised. Coding should precede reductioj. Thematazation is the process of data reduction, but coding should be first.

4. Measurements lack attitide, beleifs.....which is limitation

5. Was qualitative and quantitative parellel or sequnatial? Why? Did you use qualitative findings to develop quantitative tools? Ypu shoild have done that.

So, the authors should improve on integration of findings and qualiative methods for proper conclusion.

Reviewer #2: Dear Authors, thank you for your hard work and public health important manuscript. Although, it is a

good research output, a lot remains to be done to bring it to the expected quality both technically and

grammatically with due care before it gets to the stage of publishing as aimed.

Thus, you are strongly advised to go through all the given comments and inquiries in here with attached

PDF one by one; revise your whole manuscript to bring it to the level of expected quality both

technically and grammatically towards the positive decision

6. PLOS authors have the option to publish the peer review history of their article (what does this mean?). If published, this will include your full peer review and any attached files.

Reviewer #1: No

Reviewer #2: No

---

## [Author Response · Author response to Decision Letter 0]

19 Dec 2023

A rebuttal letter

Journal name: PLOS ONE

PONE-D-23-08920 

Title: Covid-19 vaccine uptake and its associated factors among adult population in Dangila district, Awi Zone, Northwest Ethiopia: A semi-qualitative cross sectional study

Editor’s comments Authors’ response

https://journals.plos.org/plosone/s/file?id=ba62/PLOSOne_formatting_sample_title_authors_affiliations.pdf Thank you dear editor for your valuable comment. The revised manuscript is revised for the PLOS ONE’s style requirements.

- https://doi.org/10.1177/20503121221077585

- https://doi.org/10.2147/IDR.S360103

- https://doi.org/10.3389/fpubh.2022.919494

In your revision ensure you cite all your sources (including your own works), and quote or rephrase any duplicated text outside the methods section. Further consideration is dependent on these concerns being addressed. Thank you dear editor for your comment. As per your comment, the revised manuscript has been edited for the overlapping texts. 

3. We note that Figure 1 in your submission contain map/satellite images which may be copyrighted. All PLOS content is published under the Creative Commons Attribution License (CC BY 4.0), which means that the manuscript, images, and Supporting Information files will be freely available online, and any third party is permitted to access, download, copy, distribute, and use these materials in any way, even commercially, with proper attribution. For these reasons, we cannot publish previously copyrighted maps or satellite images created using proprietary data, such as Google software (Google Maps, Street View, and Earth). For more information, see our copyright guidelines: http://journals.plos.org/plosone/s/licenses-and-copyright.

Thank you so much dear editor for your valuable comment. As per your comment, the figure has been removed from the revised manuscript for it has no a significant input for the study. 

4. We are unable to open your Supporting Information file S1 File.sav. Please kindly revise as necessary and re-upload. Thank you for the comments. The S1 File has been re-uploaded with the revised manuscript after checking for its functionality. 

Comments from Reviewer #1 Thank you for your comment. As per your comment, the necessary amendment has been made in the revised manuscript. 

Thank you this intersting piece of work on covid vaccine uptake from Ethiopia

The following key comments:

1. Integration of methods, data, findings ans conclusions is a key challenge of this work. The authors faced a great dealnof challenge to tringulate the qualitative and quantitative methodd. The findings of both methods should be mixed as appropriate instead of being put as separate pieces. So, please integrate the finding. This would pave ways to draw appropriate conclusion and and improve assimilation of findings. Thank you dear reviewer for your interesting comment. The findings of the two methods were presented sequentially, for the study was sequential mixed method. As per your comment we have refined the triangulation of the findings in the revised manuscript. 

2. Qualitative methods seem overlooked. First ly, though saturation is mentioned, there is no adequate internal evidence that saturation is reached. 1. Tentative sample sizes were not mentioned, 2. Data triangulation is not well indicated. Secondly, data analysis and rigor is not well stated. What type of thematic analysis and coding was applied- open or priori codinf? The procedure paased through transcription, coding, and analysis overly depended on one person, the principal investigator. That is not auditable, not based on peer debriefying etc. So please mention the roles of othet authors in the qualiative process. Please indicate the extent to which subjective neitrality, refxivity, tick description, etc was ensured toward dependebality of the interpretations. Please revise and improve this aspect of the study. Thank you dear reviewer for your constructive comments. As per your comments, the revised manuscript has been revised and edited for the qualitative component, and the changes have been indicated with track changes. The issues raised in this comment were addressed in page 9 of the revised manuscript and changes have been indicated by highlights. “The investigators ensured that the study was trustworthy based on Lincoln and Cuba’s criteria of credibility, dependability, conformability, and transferability. To maintain the credibility of the research findings, the study participants were observed persistently at the time of the interview. Peer-debriefing was done for the questioner. Dependability was attained through accurate documentation by minimizing spelling errors through frequently observing data and including all documents in the final report, such as including the notes written during the interview and ensuring that the details of the procedure was described to allow the readers to see the basis upon which conclusions were made. To achieve conformability of the study findings, raw data, interview and observation notes, documents and records collected from the field, and others were documented for cross checking. To maintain the transferability of the finding, appropriate probes was used to obtain detailed information on responses, and study participants were selected based on their specific purpose to answer study questions and to get greater in-depth findings”

3. In the abstract you mentioned..." trascribed reduced, coded, ......". This should be revised. Coding should precede reductioj. Thematazation is the process of data reduction, but coding should be first.

 Thank you dear reviewer for your constructive comments. As per your comment, the necessary amendment has been made in the revised manuscript. 

4. Measurements lack attitide, beleifs.....which is limitation Thank you dear reviewer for your constructive comments. The attitude and knowledge of respondents were assessed in the quantitative component of the study and the findings were presented both in text and in table 3, and located in page 12 of the revised manuscript. 

5. Was qualitative and quantitative parellel or sequnatial? Why? Did you use qualitative findings to develop quantitative tools? Ypu shoild have done that. So, the authors should improve on integration of findings and qualiative methods for proper conclusion. Thank you dear reviewer for your constructive comments. The study was sequential mixed method. In this case we analyzed the qualitative part after the quantitative analysis was made. Hence, the revised manuscript has been edited in that way.

Comments from Reviewer #2 

Dear Authors, thank you for your hard work and public health important manuscript. Although, it is a

good research output, a lot remains to be done to bring it to the expected quality both technically and

grammatically with due care before it gets to the stage of publishing as aimed.

Thus, you are strongly advised to go through all the given comments and inquiries in here with attached

PDF one by one; revise your whole manuscript to bring it to the level of expected quality both

technically and grammatically towards the positive decision

 Thank you dear reviewer for your constructive comments. The revised manuscript has been edited for technical and grammatical errors. Besides, all comments of the reviewers and editors have been corrected and indicated by track changes in the revised manuscript.

---

## [Decision Letter · Decision Letter 1]

24 Jan 2024

PONE-D-23-08920R1Covid-19 vaccine uptake and its associated factors among adult population in Dangila district, Awi Zone, Northwest Ethiopia: A semi-qualitative cross sectional studyPLOS ONE

Dear Dr. Anley,

Thank you for submitting your manuscript to PLOS ONE. After careful consideration, we feel that it has merit but does not fully meet PLOS ONE’s publication criteria as it currently stands. Therefore, we invite you to submit a revised version of the manuscript that addresses the points raised during the review process.

We look forward to receiving your revised manuscript.

Kind regards,

Mesfin Gebrehiwot Damtew (PhD)

Academic Editor

PLOS ONE

Journal Requirements:

Reviewers' comments:

Reviewer's Responses to Questions

**Comments to the Author**

1. If the authors have adequately addressed your comments raised in a previous round of review and you feel that this manuscript is now acceptable for publication, you may indicate that here to bypass the “Comments to the Author” section, enter your conflict of interest statement in the “Confidential to Editor” section, and submit your "Accept" recommendation.

Reviewer #2: (No Response)

2. Is the manuscript technically sound, and do the data support the conclusions?

Reviewer #2: Partly

3. Has the statistical analysis been performed appropriately and rigorously? 

Reviewer #2: I Don't Know

4. Have the authors made all data underlying the findings in their manuscript fully available?

Reviewer #2: Yes

5. Is the manuscript presented in an intelligible fashion and written in standard English?

Reviewer #2: Yes

6. Review Comments to the Author

Reviewer #2: Dear Author , thank you for sending your improved version of the manuscript on COVID-19 Vaccine up take and associated factors . Although you tried to address many comments forwarded from the reviewers ,still the problem of addressing clearly the objective and expected contents of a mixed methods approach is not clearly addressed . 1.In addition, in your title section you said it was a <<semi qualitative="">>study design and in methodology section you said it is a mixed method design . Thus, it is good to be consistent . 2. Most importantly in your discussion section the concept of triangulation or support of quantitative data with that of qualitative is still not written to the expected standard or quality . 3. Be it sequential or not , I suppose, it should be clearly articulated or shown in such a way that whether a finding of a method was supported or not by the other data, source or method during triangulation for better validity of the finding/s rather than giving each of them in independent sections</semi>

7. PLOS authors have the option to publish the peer review history of their article (what does this mean?). If published, this will include your full peer review and any attached files.

Reviewer #2: No

---

## [Author Response · Author response to Decision Letter 1]

8 Feb 2024

A rebuttal letter 2

Journal name: PLOS ONE

PONE-D-23-08920 

Title: Covid-19 vaccine uptake and its associated factors among adult population in Dangila district, Awi Zone, Northwest Ethiopia: A mixed method study

Editor’s comments Authors’ response

1. Please review your reference list to ensure that it is complete and correct. If you have cited papers that have been retracted, please include the rationale for doing so in the manuscript text, or remove these references and replace them with relevant current references. Any changes to the reference list should be mentioned in the rebuttal letter that accompanies your revised manuscript. If you need to cite a retracted article, indicate the article’s retracted status in the References list and also include a citation and full reference for the retraction notice. Thank you dear editor for your valuable comment. The revised manuscript is revised for the presence of retracted references. We found that none of them are retracted. 

Comments from Reviewer #2 

Dear Author, thank you for sending your improved version of the manuscript on COVID-19 Vaccine up take and associated factors . Although you tried to address many comments forwarded from the reviewers ,still the problem of addressing clearly the objective and expected contents of a mixed methods approach is not clearly addressed

1. In addition, in your title section you said it was a <>study design and in methodology section you said it is a mixed method design. Thus, it is good to be consistent. Thank you dear reviewer for your constructive comments. As per your comment, the revised manuscript is edited for consistency in terms of terminologies in the design both in the title and methodology. 

2. Most importantly in your discussion section the concept of triangulation or support of quantitative data with that of qualitative is still not written to the expected standard or quality . Thank you dear reviewer for your constructive comments. As per your comment we have included the following paragraph in the discussion section of the revised manuscript. The detail of the qualitative finding analysed sequentially has been written in the result part. Methdological aspects have been also written in the methodology part.

“The quantitative findings were further validated through a meticulous analysis of the qualitative data. Thematic exploration concerning barriers to COVID-19 vaccine acceptance among the adult population in Dangila district, Awi Zone, revealed three primary themes, each with seven sub-themes. These included Vaccine-related factors, Personal factors, and Environmental factors, encompassing issues such as the Availability of vaccine, Misperceptions about vaccine efficacy and safety, Lack of knowledge about the COVID-19 vaccine, Mis trust, Fear of adverse effects, Social media influence, and Religious beliefs. Notably, these qualitative insights mirror the patterns observed in the quantitative analysis, affirming the robustness and consistency of the findings across both methodologies.”

3. Be it sequential or not , I suppose, it should be clearly articulated or shown in such a way that whether a finding of a method was supported or not by the other data, source or method during triangulation for better validity of the finding/s rather than giving each of them in independent sections. Yes dear reviewer, we totally agreed with this and hence, we have stated this triangulation and consistency of findings of the two methods in the discussion part and it has been indicated in the revised manuscript with track changes. Thank you for your efforts to improve our manuscript.

---

## [Decision Letter · Decision Letter 2]

27 Feb 2024

PONE-D-23-08920R2Covid-19 vaccine uptake and its associated factors among adult population in Dangila district, Awi Zone, Northwest Ethiopia: A mixed method studyPLOS ONE

Dear Dr. Anley,

Thank you for submitting your manuscript to PLOS ONE. After careful consideration, we feel that it has merit but does not fully meet PLOS ONE’s publication criteria as it currently stands. Therefore, we invite you to submit a revised version of the manuscript that addresses the points raised during the review process.

We look forward to receiving your revised manuscript.

Kind regards,

Mesfin Gebrehiwot Damtew (PhD)

Academic Editor

PLOS ONE

Journal Requirements:

Reviewers' comments:

Reviewer's Responses to Questions

**Comments to the Author**

1. If the authors have adequately addressed your comments raised in a previous round of review and you feel that this manuscript is now acceptable for publication, you may indicate that here to bypass the “Comments to the Author” section, enter your conflict of interest statement in the “Confidential to Editor” section, and submit your "Accept" recommendation.

Reviewer #1: (No Response)

Reviewer #2: All comments have been addressed

2. Is the manuscript technically sound, and do the data support the conclusions?

Reviewer #1: Partly

Reviewer #2: Yes

3. Has the statistical analysis been performed appropriately and rigorously? 

Reviewer #1: No

Reviewer #2: I Don't Know

4. Have the authors made all data underlying the findings in their manuscript fully available?

Reviewer #1: (No Response)

Reviewer #2: Yes

5. Is the manuscript presented in an intelligible fashion and written in standard English?

Reviewer #1: No

Reviewer #2: Yes

6. Review Comments to the Author

Reviewer #1: Dear authors and editor,

Thank you for inviting me to review this interesting work. I will provide my general remark on the manuscript as follows:

1. I know that it is not necessary to publish on PLOS ONE when a study has only few finding. This study has not added to existing literature, even in Ethiopia.

2. It is great that this is a mixed-method study. I would suggest the authors to specifically mention which type of mixed-method is this? Is it convergent/parallel or explanatory sequential or exploratory sequential one? The problem, such a study should have benefited a lot from exploratory sequential mixed-method. There may be many local and cultural beliefs and influences specific to Dangila that the authors could have explored qualitatively first then provide quantitative evidence latter (this can be a convincing research question to be asked at this time).

3. Again another problem exists analysis approach of this mixed-method. As it currently stands, the authors did not triangulate the quantitative and qualitative findings. Each finding appeared stand alone- this means the findings are not assimilated well and not ready for discussion. Please revise. Don't have knowledge, attitude....titles twice in this same manuscript. If you start describing or explaining about knowledge about COVID-19 or its vaccine, please triangulate both qualitative and quantitative findings in that same sub-section. You should not come again explaining about knowledge qualitatively or quantitatively. Same holds true with other reported sub-titles such as attitude or social influences.

4. Please say "religious beliefs" instead of "religious believe". please check such issues throughout the manuscript. You may use freely available softwares such as Grammarly or other support centers.

5. Measurement and analysis of knowledge is not exhaustive and convincing.

6. Using mean to categorize knowledge and or attitude into poor and good is unnecessary nomination or judgement of the respondents- use standard reference to category these. Our use quartiles or standardized mean to report in the form of proportion or use count of basic knowable knowledge items by referring to WHO or national expectations. You can reply on expert discussion or opinion to identify which are the minimum knowable items, otherwise.

7. Before moving on to discussion, please make proper triangulation of both qualitative and quantitative data

8. Qualitative findings are shallow. please immerse yourself with the data and provide tick description.

9. Provide has detail techniques through which you assured credibility, dependability, transferability and confirmability of qualitative findings.

I would recommend the authors, to critically revise this work.

Regards

Reviewer #2: (No Response)

7. PLOS authors have the option to publish the peer review history of their article (what does this mean?). If published, this will include your full peer review and any attached files.

Reviewer #1: No

Reviewer #2: **Yes: **Takele Menna Adilo

---

## [Author Response · Author response to Decision Letter 2]

5 Mar 2024

A rebuttal letter 3

Journal name: PLOS ONE

PONE-D-23-08920 

Title: Covid-19 vaccine uptake and its associated factors among adult population in Dangila district, Awi Zone, Northwest Ethiopia: A mixed method study

Editor’s comments Authors’ response

Journal Requirements:

Please review your reference list to ensure that it is complete and correct. If you have cited papers that have been retracted, please include the rationale for doing so in the manuscript text, or remove these references and replace them with relevant current references. Any changes to the reference list should be mentioned in the rebuttal letter that accompanies your revised manuscript. If you need to cite a retracted article, indicate the article’s retracted status in the References list and also include a citation and full reference for the retraction notice. Thank you dear editor for your valuable comment. The revised manuscript is revised for the presence of retracted references. We found that none of them are retracted. 

Comments from reviewer #1

Thank you for inviting me to review this interesting work. I will provide my general remark on the manuscript as follows:

1. I know that it is not necessary to publish on PLOS ONE when a study has only few finding. This study has not added to existing literature, even in Ethiopia.

2. It is great that this is a mixed-method study. I would suggest the authors to specifically mention which type of mixed-method is this? Is it convergent/parallel or explanatory sequential or exploratory sequential one? The problem, such a study should have benefited a lot from exploratory sequential mixed-method. There may be many local and cultural beliefs and influences specific to Dangila that the authors could have explored qualitatively first then provide quantitative evidence latter (this can be a convincing research question to be asked at this time). Absolutely, I completely understand your perspective. While this study may not seem groundbreaking at first glance, it actually addresses a crucial issue: the low COVID-19 vaccine uptake in the region. By employing a mixed-method, community-based approach, this study likely offers valuable insights and potential strategies for improving vaccine acceptance and uptake. Despite its seemingly limited scope, its potential impact on public health in Ethiopia cannot be overlooked. 

The mixed method study employed was an explanatory sequential mixed-method. We preferred this approach to clarify or elaborate on quantitative findings by exploring participants' perspectives or contexts.

3. Again another problem exists analysis approach of this mixed-method. As it currently stands, the authors did not triangulate the quantitative and qualitative findings. Each finding appeared stand-alone- this means the findings are not assimilated well and not ready for discussion. Please revise. Don't have knowledge, attitude....titles twice in this same manuscript. If you start describing or explaining about knowledge about COVID-19 or its vaccine, please triangulate both qualitative and quantitative findings in that same sub-section. You should not come again explaining about knowledge qualitatively or quantitatively. Same holds true with other reported sub-titles such as attitude or social influences. Thank you dear reviewer for your constructive comments. As per your suggestion, the revised manuscript is writing the quantitative and qualitative findings together with proper triangulation. The changes made have been shown with track changes in the revised manuscript. 

4. Please say "religious beliefs" instead of "religious believe". please check such issues throughout the manuscript. You may use freely available softwares such as Grammarly or other support centers. Thank you dear reviewer for your comment. The necessary correction has been made in the revised manuscript. 

5. Measurement and analysis of knowledge is not exhaustive and convincing. 

6. Using mean to categorize knowledge and or attitude into poor and good is unnecessary nomination or judgement of the respondents- use standard reference to category these. Our use quartiles or standardized mean to report in the form of proportion or use count of basic knowable knowledge items by referring to WHO or national expectations. You can reply on expert discussion or opinion to identify which are the minimum knowable items, otherwise.

 Thank you for your constructive comments. The measurement of knowledge and attitude has been indicated in the methodology section of the revised manuscript with citation of the appropriate reference. 

7. Before moving on to discussion, please make proper triangulation of both qualitative and quantitative data

 Thank you dear reviewer for your constructive comments. As per your comment, the necessary correction has been made in the revised manuscript.

8. Qualitative findings are shallow. Please immerse yourself with the data and provide tick description. Thank you for your comments. Some modifications have been made in the revised manuscript.

9. Provide has detail techniques through which you assured credibility, dependability, transferability and confirmability of qualitative findings. Thank you for your constructive comment. We included the following paragraph in the methodology section. “We ensured that the study was trustworthy based on Lincoln and Cuba’s criteria of credibility, dependability, conformability, and transferability. To maintain the credibility of the research findings, the study participants were observed persistently at the time of the interview. Peer-debriefing was done for the questioner. Dependability was attained through accurate documentation by minimizing spelling errors through frequently observing data and including all documents in the final report, such as including the notes written during the interview and ensuring that the details of the procedure was described to allow the readers to see the basis upon which conclusions were made. To achieve conformability of the study findings, raw data, interview and observation notes, documents and records collected from the field, and others were documented for cross checking. To maintain the transferability of the finding, appropriate probes were used to obtain detailed information on responses, and study participants were selected based on their specific purpose to answer study questions and to get greater in-depth findings”. 

Comments from Reviewer #2 

All comments have been addressed. Thank you so much dear reviewer for your valuable input for the improvement of our manuscript.

---

## [Decision Letter · Decision Letter 3]

15 Mar 2024

PONE-D-23-08920R3Covid-19 vaccine uptake and its associated factors among adult population in Dangila district, Awi Zone, Northwest Ethiopia: A mixed method studyPLOS ONE

Dear Dr. Anley,

Thank you for submitting your manuscript to PLOS ONE. After careful consideration, we feel that it has merit but does not fully meet PLOS ONE’s publication criteria as it currently stands. Therefore, we invite you to submit a revised version of the manuscript that addresses the points raised during the review process.

We look forward to receiving your revised manuscript.

Kind regards,

Mesfin Gebrehiwot Damtew (PhD)

Academic Editor

PLOS ONE

Journal Requirements:

Reviewers' comments:

Reviewer's Responses to Questions

**Comments to the Author**

1. If the authors have adequately addressed your comments raised in a previous round of review and you feel that this manuscript is now acceptable for publication, you may indicate that here to bypass the “Comments to the Author” section, enter your conflict of interest statement in the “Confidential to Editor” section, and submit your "Accept" recommendation.

Reviewer #1: (No Response)

2. Is the manuscript technically sound, and do the data support the conclusions?

Reviewer #1: (No Response)

3. Has the statistical analysis been performed appropriately and rigorously? 

Reviewer #1: (No Response)

4. Have the authors made all data underlying the findings in their manuscript fully available?

Reviewer #1: (No Response)

5. Is the manuscript presented in an intelligible fashion and written in standard English?

Reviewer #1: (No Response)

6. Review Comments to the Author

Reviewer #1: Authors are suggested to annex item by item description of knowledge and attitude questions so that readers would compare the cumulative reports in the respective tables of the result section. This same figures of knowledge and attitude reported after changing the analysis approach between revised version 2 and 3. This can be trusted when the item by item reportes are annexed. The second, comment is that the triangulations are not adequately done. Please pick beleifs and perceptions in the qualitative and triangulate them with quantitative knowledge and attitude sections. Do not report qualitative findings seeminly as separate section of the study.

Regards

7. PLOS authors have the option to publish the peer review history of their article (what does this mean?). If published, this will include your full peer review and any attached files.

Reviewer #1: No

---

## [Author Response · Author response to Decision Letter 3]

2 Apr 2024

A rebuttal letter 4

Journal name: PLOS ONE

PONE-D-23-08920 

Title: Covid-19 vaccine uptake and its associated factors among adult population in Dangila district, Awi Zone, Northwest Ethiopia: A mixed method study

Editor’s comments Authors’ response

Journal Requirements:

Please review your reference list to ensure that it is complete and correct. If you have cited papers that have been retracted, please include the rationale for doing so in the manuscript text, or remove these references and replace them with relevant current references. Any changes to the reference list should be mentioned in the rebuttal letter that accompanies your revised manuscript. If you need to cite a retracted article, indicate the article’s retracted status in the References list and also include a citation and full reference for the retraction notice. Thank you dear editor for your valuable comment. The revised manuscript is revised for the presence of retracted references. We found that none of them are retracted. 

Comments from reviewer #1

1. Authors are suggested to annex item by item description of knowledge and attitude questions so that readers would compare the cumulative reports in the respective tables of the result section. This same figures of knowledge and attitude reported after changing the analysis approach between revised version 2 and 3. This can be trusted when the item by item reportes are annexed. The second, comment is that the triangulations are not adequately done. Please pick beleifs and perceptions in the qualitative and triangulate them with quantitative knowledge and attitude sections. Do not report qualitative findings seeminly as separate section of the study.

 � Thank you dear reviewer for your constructive comments which are too crucial for the improvement of our manuscript. 

As per your comment, the item by item descriptions of knowledge and attitude questions have been uploaded as supplementary information file with the revised manuscript. 

Regarding to the triangulation of the qualitative finding with quantitative one, we have adhered to your valuable comments. Beleifs and perceptions in the qualitative finding are not resented separately. We have triangulated it in the heading “Factors associated with vaccine uptake” (page 14, line 351-401). Here, factors identified by quantitative analysis have been triangulated with qualitative findings where beleifs and perceptions have been triangulated too. 

The Respondents’ Knowledge and Attitude characteristics described at (page 12, line 315-323) is a purely descriptive finding where the qualitative findings of related terms shouldn’t be triangulated with.

---

## [Editor Report · Decision Letter 4]

9 Apr 2024

Covid-19 vaccine uptake and its associated factors among adult population in Dangila district, Awi Zone, Northwest Ethiopia: A mixed method study

PONE-D-23-08920R4

Dear Dr. Anley,

We’re pleased to inform you that your manuscript has been judged scientifically suitable for publication and will be formally accepted for publication once it meets all outstanding technical requirements.

Kind regards,

Mesfin Gebrehiwot Damtew (PhD)

Academic Editor

PLOS ONE